# Anatolian Short-Horned Grasshoppers Unveiled: Integrating Biogeography and Pest Potential

**DOI:** 10.3390/insects15010055

**Published:** 2024-01-12

**Authors:** Battal Çıplak, Onur Uluar

**Affiliations:** Department of Biology, Faculty of Science, Akdeniz University, 07058 Antalya, Turkey; onuruluar@gmail.com

**Keywords:** Anatolia, Orthoptera, Caelifera, biogeography, pest management, ecological-niche modelling

## Abstract

**Simple Summary:**

Short-horned grasshoppers (Orthoptera; Caelifera) in Anatolia inhabit diverse habitats, aligning with the topographical and climatological heterogeneity of the region. In addition to certain swarming species, attention must be given to the pest potential of several pullulating species within the Anatolian fauna. This study seeks to classify Anatolia’s short-horned grasshoppers from a biogeographical perspective and integrate these data to comprehend the future pest potential of non-swarming species, especially in the context of climate change. Our results reveal the following: (i) Acrididae and Pamphagidae are the most diverse families in Anatolia; (ii) approximately 40% of Caelifera and 71% of Pamphagidae are endemic, marking Anatolia as a biodiversity hotspot; (iii) the phytogeographical order of four provinces based on Caelifera diversity is Irano-Anatolia, Euro-Siberia, Mediterranean, and Mesopotamia; and (iv) based on our ecological modelling and personal observations, *Dociostaurus maroccanus*, *Locusta migratoria*, *Calliptamus italicus*, *Heteracris pterosticha*, *Notostaurus anatolicus*, *Oedipoda miniata*, and *O. schochii* should be monitored due to their pest potential.

**Abstract:**

Biogeographically, Anatolia harbours a rich diversity of short-horned grasshoppers (Orthoptera, Caelifera). The number of species recorded from Anatolia so far stands at 300. They inhabit diverse habitats ranging from arid Eremial to Euro-Siberian-like montane meadows, aligning with the topographical and climatological heterogeneity of Anatolia. Alongside some swarming species, the pest potential of several pullulating species needs attention. This is especially important concerning global warming, a scenario expected to be more severe in the Northern Mediterranean Basin in general and Anatolia specifically. A faunal list of biogeographic Anatolia, the area extending from the Aegean Sea in the west to the intermountain basin of the Caucasus in the northeast, the lowlands of Lake Urmia in the east, and Mesopotamia in the southeast, was developed. The recorded species were classified according to the phytogeographical provinces of Anatolia. Distributions of the species with the potential for pullulating were modelled using ecological-niche-modelling approaches for the present and future. The results have the potential to lead to the development of a concept that merges biogeography and the pest potential of certain Anatolian grasshopper species. Our results reveal the following: (i) Acrididae and Pamphagidae are the most diverse families represented in Anatolia; (ii) roughly 40% of Caelifera and 71% of Pamphagidae are endemics, suggesting Anatolia is a biodiversity hotspot; (iii) according to Caelifera diversity, the phytogeographical provinces of Anatolia follow an order of Irano-Anatolia, Euro-Siberia, Mediterranean, and Mesopotamia; and (iv) based on ecological modelling and personal observations, *Dociostaurus maroccanus*, *Locusta migratoria*, *Calliptamus italicus*, *Heteracris pterosticha*, *Notostaurus anatolicus*, *Oedipoda miniata*, and *O. schochii* should be monitored regarding their pest potential.

## 1. Introduction

The traces of data related to Anatolian Orthoptera in general and Caelifera specifically can be found in publications on European Orthoptera from as far back as the late 19th century. The earliest publications specifically dealing with Anatolian Orthoptera appeared at the end of the 19th century [1] and the beginning of the 20th century [2,3,4,5,6]. A new era began with the First World War, especially following the studies by Boris P. Uvarov [7,8,9]. Later studies by Uvarov [10] provided a significant contribution to the determination of Anatolian grasshopper fauna. Simultaneously, Ebner [11,12] and Ramme [13,14,15,16,17] conducted extensive studies on orthopteroid insects in this region. By the 1950s, the list of Anatolian grasshoppers was almost complete. Ramme [13] listed 157 species/subspecies of Caelifera from Anatolia. The study by Bei-Bienko and Mistshenko [18] is another publication containing comprehensive data about Anatolian Caelifera. T. Karabağ was the first local orthopterist who specifically prepared a catalogue for Orthoptera of Turkey [19], in which he listed 206 species/subspecies of short-horned grasshoppers. Weidner [20] specifically reviewed Caelifera from Turkey and listed 187 species (206 species/subspecies); soon after, this number increased to 229 as per the study by Demirsoy [21]. Currently, the number of taxa (species/subspecies) belonging to Caelifera in Turkey run into 300 species (244 according to Çıplak et al. [22] and 288 according to Ünal [23]). Although there are taxonomical uncertainties for some taxa, the current picture of Anatolian short-horned grasshopper fauna is more or less clear and allows us to draw general conclusions about their ecology, biogeography, evolution, and pest potential. 

Orthoptera were considered a marker group in defining the biogeography of Anatolia, the area extending from the Aegean Sea in the west to the intermountain basin of the Caucasus in the northeast, the lowlands of Lake Urmia in the east, and Mesopotamia in the southeast [24]. This is not surprising, as one of the earliest and preliminary publications on Anatolian biogeography focused on the distribution of Orthoptera [7]. Uvarov’s study constituted the basis for subsequent studies [13,20,21]. Recently, a significant number of publications on Anatolian biogeography containing Orthoptera content have emerged [25,26,27]. The coupling of Orthoptera and Anatolian biogeography has peculiarities for several reasons. First, compared to other invertebrate groups, the Orthoptera fauna of Anatolia is relatively well known, a consequence of data accumulation since the 19th century. Second, Orthoptera is a diverse lineage in Anatolia, including sublineages with different ecological preferences, and the diversity of ecological preferences of the sublineages correlates with the eco-geographic fragmentation of Anatolia. Thus, sublineages belonging to Orthoptera have the potential to serve as model groups for addressing questions related to Anatolian biogeography. Third, several orthopteran lineages possess imprints of the tempestuously dynamic geographic history of Anatolia in their phylogeny. Connected to this radiation history, Anatolia harbours a considerable number of tribal, generic, or species taxa that are endemic or predominantly Anatolian in distribution [21,25,26,27,28,29]. Thus, studies on the biophylogeography of Anatolian Orthoptera provide a multidimensional perspective, extending from taxonomy to evolution and ecology. 

The first classification of Anatolian orthopteroid species/genera according to their eco-geographic preferences was provided by Uvarov [7]. He applied a system of four eco-geographic subregions of the Palaearctic region (namely, Boreal, Steppe, Mediterranean, and Eremian) to Western Asia (Anatolia, Caucasus, and Northern Iran) (Figure 1A). Regarding Anatolia, Uvarov [7] reported the existence of representatives from all four eco-geographic sections, but those of the Mediterranean and Eremian were dominant. The ecogeographic classification of Anatolian Caelifera by Weidner [20] (see Figure 1B), also followed by Demirsoy [21], was largely different from that developed by Uvarov [7], not only in terms of the names of eco-geographic sections but also in terms of their borders and typical representative species. The main differences are as follows: (i) Uvarov proposed the Black Sea Basin in the Mediterranean subregion, while Weidner placed it in the Siberian or Steppe subregion; (ii) Uvarov suggested the Levantine extension to Anatolia as Syrian Anatolia in the Mediterranean, while Weidner [4] considered it a part of the Afro-Eremian subregion; (iii) Uvarov distinguished the Mediterranean section by naming it Anatolio-Balkan fauna, while Weidner did not; and (iv) Weidner defined several refugial areas in Anatolia, while Uvarov identified none. Further differences can also be noted by comparing both publications [7,20] (compare Figure 1A,B).

The above-mentioned preliminary studies, which were followed by many subsequent publications, indicate the necessity of a definition considering the vegetation of the area and pose significant questions to be answered. First, Anatolia is highly complex in its geography and climate, so choosing a criteria for defining habitat content, and thus the application of any general classification, remain too simplistic. Second, species with a particular ecological preference may penetrate different eco-geographic sections due to the presence of island-like refugial areas, and this hinders the definition of faunal elements that are typical for a section. Third, Anatolia harbours a considerable percentage of endemic species [7,13,20], and a proper eco-biogeographic definition of the region requires considering its own features, such as vegetation [30]. Fourth, all previous studies [7,20,21] adopted an eco-biogeographic perspective, considering specific elements to have arrived in Anatolia from somewhere outside this region. Such a perspective is misleading phylogeographically, as are several lineages that specifically originated and evolved here, such as several genera of Pamphagidae [31,32], some lineages of Gomphocerinae [33], and many lineages of Ensifera [9,12,24]. As suggested by both early [34] and recent studies [9,25,27,35], the reverse case, i.e., defining Anatolia as a centre of radiation and dispersing from Anatolia to the surrounding geographic area, seems much more likely. Documenting all incompatible or inadequate accounts on Anatolian biogeography is beyond the aims of the present study, but all indicate the necessity of a reconsideration, particularly with respect to the distribution pattern of Caelifera diversity.

Some of the swarming Caeliferan locust species occur in Anatolia. The desert locust, *Schistocerca gregaria*, is the best-known species, but Anatolia remains outside of its recession range [36,37,38,39], and there have been no desert locust swarms in the region since the 1960s [40]. The other three outbreaking species that have caused serious damage in the past and still have the potential to inflict damage in the area are the Moroccan locust, *Dociostaurus maroccanus* [41,42]; the migratory locust, *Locusta migratoria*; and the Italian locust, *Calliptamus italicus* [43,44]. The assessment of their potential in the context of global warming seems of particular importance [40]. Additionally, there have been occasional and localized outbreaks of grasshopper species, such as *Heteracris pterosthica*, *Notostaurus anatolicus*, *Arcyptera labiata*, and *Calliptamus* spp., aside from *C. italicus* (namely, *C. barbarus* and *C. tenuicercis*), in Anatolia [44,45,46]. Furthermore, based on the experience of the first author, certain species (e.g., *Chorthippus* spp. and *Oedipoda* spp.) proliferate regionally and they have caused damage in certain years. The continuance of global warming may change habitat characteristics and disturb species presence, consequently leading to shifts in their distribution areas or phenology/life history characteristics or even driving them to extinction [26]. Aridification is the most probable consequence of global warming, especially in the Eastern Mediterranean Basin, including Anatolia [47]. Aridification may lead to the expansion of the Eremian or arid eco-zones, and such expansion may provide opportunities for species with Eremian habitat preferences to expand their ranges or even proliferate and become pests in large parts of Anatolia, excluding the sea basin zones. Testing this probability is of special importance and may provide a corridor between biogeography and pest potential estimation.

The present study is intended to provide a perspective for merging biogeography and pest potential estimation for short-horned grasshoppers in Anatolia. This aim will be achieved by (1) providing a faunistic list of Caelifera, (2) defining species or supra-species lineage eco-biogeographic characteristics in reference to Anatolian climatic fragments and phytogeographic provinces (as all members of the suborder are herbivorous and some are oligophagous), and (3) estimating the future pest potential of pullulating species via modelling the distribution of species with pest potential.

## 2. Materials and Methods

This study was planned in three successive modules, with the first aimed at providing an updated checklist for Anatolian short-horned grasshoppers. Previous checklists, mainly those presented in [13,20,21,31], as well as recent ones [22,23], were considered as a starting point. Species/subspecies from these publications were adopted to establish new lists, and the taxa were cross-checked against Orthoptera Species File 2 (OSF2) [48] for nomenclatural changes and taxonomic clarification. OSF2 [48] was also utilized to determine publications related to each taxon. Taxonomic/faunistic publications were examined to determine the intra/extra-Anatolian distribution of each species/subspecies.

The second module of the study involves classifying Anatolian short-horned grasshoppers according to their eco-geographic preferences. The eco-geographic preferences of the species/subspecies were classified according to the phytogeographic provinces of Anatolia defined by Zohary [30], consisting of four sections: Mediterranean, Euro-Siberian, Irano-Anatolian, and Mesopotamia (see also [25,49]) (Figure 2). This classification was deemed reasonable considering that locusts and short-horned grasshoppers are herbivorous insects, thus leading to the expectation of a coupling between plant and grasshopper compositions. Although this classification partly corresponds to that developed by Uvarov [7] or Weidner [4], as evidenced by, for example, the consideration of the Mediterranean Region, which is common to all, the sections considered here are different, at least with respect to the intra-Anatolian borders. The species list was prepared as a table indicating species presence/absence per section in Anatolia. The endemic taxa were also identified in the table. This table was used to infer the habitat preferences of species/subspecies, calculate section diversity, and derive the general pattern of the diversity characteristics of Anatolian short-horned grasshoppers.

The third module of this study focuses on pest species or species with pest potential, particularly considering climate-warming scenarios. *Dociostaurus moroccanus*, *Locusta migratoria*, and *Calliptamus italicus* are recognized as outbreaking species in the region [36,37,38,39,40,41,42]. In addition to these three species, *Heteracris pterosthica*, *Dociostaurus brevicollis*, *Notostaurus anatolicus*, *Arcyptera labiata*, *C. barbarus*, and *C. tenuicercis* have been reported to be occasionally outbreaking species in Anatolia in arid and semi-arid areas [40,43,44,45,46]. Furthermore, based on the experience of the first author, *Chorthippus dichrous*, *Ch. karelinii*, and *Euchorthippus pulvinatus* were identified as pullulating species in highland meadows, and so were *Oedipoda miniata* and *O. schochii* in arid areas. Current and future distributions of these species were estimated using species distribution modelling. Current and future species distribution predictions were conducted via the *raster* [50] and *sdm* [51] packages in the *R environment* [52] for the 14 species of Acrididae with pest potential in Anatolia. The species’ occurrence data (Appendix A) were gathered from various sources [13,19,20,21,53,54,55,56,57,58,59,60,61,62,63,64,65,66,67], with the majority of localities coming from samples preserved in the author’s personal collection at AUZM (Akdeniz University Zoology Museum, Antalya, Turkey) and MEVBIL (Molecular Evolution and Biogeography Lab.) at Akdeniz University. Publications containing records of the listed species were also cross-referenced. 

Bioclimatic data for the near present (1970–2000) and future (2061–2080 average, 2070, CCSM4, RCP 8.5) were downloaded from the WorldClim database v.2 [68] at a spatial resolution of 2.5 min (~4.5 km^2^) for modeling. Variance inflation factor (VIF) scores were calculated to exclude collinear bioclimatic variables, and uncorrelated variables were used for the modeling distribution of each species. Pseudo-absence points were created using the *gRandom* method (n = 1000) by means of various prediction models, including Generalized Additive Models (GAM), Generalized Linear Models (GLM), and Maximum Entropy (MaxEnt). The subsampling test percentage and number of replicates were set at 10 and 20, respectively. Model performance parameters, AUC (Area Under the Curve [69]), and TSS (the true skill statistics [70]) were calculated for each model, and the consensus predictions of each model were used through a “weighted” scheme [51].

## 3. Results

Two hundred and eighty-four species of Caelifera, encompassing 79 genera, have been documented in Anatolia (Appendix B Table A1). The most diverse family in Anatolia is Acrididae, with a total of 175 species representing 57 genera classified under eight subfamilies. Pamphagidae occupies the second position, with a total of 91 species representing 15 genera classified under two subfamilies. Tetrigidae takes the third position with eight species from two different genera of the nominate subfamily. The remaining three families are represented by a few species in Anatolia. Tridactylidae consists of four species representing three genera from two different subfamilies, while Pyrgomorphidae and Dericorythidae each have three species of a single genus (Table 1; Figure 3).

Endemic species account for more than 39.4% of Anatolian Caelifera, with 112 out of the total 284 being endemic. Among these, 65 out of the total 91 species (71.4%) belong to Pamphagidae, and 47 out of the total 175 species (15%) belong to Acrididae (Table 1, Figure 3). The diversity of the other four families is limited, with Tridactylidae, Tetrigidae, Pyrgomorphidae, and Dericorythidae each having fewer than 10 species and no endemic representatives in Anatolia (Table 1, Figure 2). The genera *Ebnerodes*, *Glyphothmethis*, *Paranocarodes*, *Paranothrotes*, *Pseudonothrotes*, *Nocarodes*, *Nocaracris*, and *Prionosthenus*, all within Pamphagidae, are either endemic or predominantly Anatolian in distribution. Although Acrididae is the most diverse family, only the monotypic genera *Rammepodisma* and *Demirsoyus* are endemic, and there are no polytypic genera that are endemic or predominantly Anatolian in distribution (Appendix B Table A1).

Each of the four provinces exhibits a different Caelifera faunal composition (Appendix B Table A1, Table 2, Figure 3). The most diverse province is Irano-Anatolia, with 193 species (68% out of the 284 species recorded from Turkey), including 121 from Acrididae, 59 from Pamphagidae, and 13 from the remaining four families. The second-most-diverse province is Euro-Siberia, with 131 species (46% of the total), comprising 99 from Acrididae, 24 from Pamphagidae, 7 from Tetrigidae, and 1 from Tridactylidae (Appendix B Table A1, Table 2, and Figure 4). The third-most-diverse province is the Mediterranean province, with 127 species (45% of the total), including 89 from Acrididae, 28 from Pamphagidae, and the remaining 10 from Tridactylidae, Tetrigidae, and Pyrgomorphidae. Mesopotamia is the least diverse province, with 55 species (19% of the total). Acrididae and Pamphagidae are the two most diverse families in all four provinces, as for entire Anatolia. According to regional diversity, the richest province is Irano-Anatolia, and the poorest is Mesopotamia for both Acrididae and Pamphagidae. Pyrgomorphidae is absent in Euro-Siberia, Tridactylidae is absent in Mesopotamia, and Dericorythidae is absent in the Mediterranean and Euro-Siberian provinces (Appendix B Table A1, Figure 4).

Variance inflation factor (VIF) scores per bioclimatic factor indicated that the number of retained bioclimatic variables per species was six for *Dociostaurus maroccanus* and *Heteracris pterosticha*; seven for *Arcyptera labiata*, *Calliptamus italicus*, *C. barbarous*, *C. tenuicercis*, *Dociostaurus brevicollis*, *Locusta migratoria*, and *Oedipoda miniata* and eight for *Chorthippus dichrous*, *Ch. karelini*, *Euchorthippus pulvinatus*, *O. schochii*, and *Notostaurus anatolicus* (Table 3). Of the 19 bioclimatic variables, the maximum temperature of the warmest month (BIO5), the minimum temperature of the coldest month (BIO6), the minimum temperature of the coldest quarter (BIO11), precipitation in the driest quarter (BIO17), and precipitation in the coldest quarter (BIO19) were uninformative (correlated) for all 14 species. Temperature of annual range (BIO7; BIO5/BIO6) and precipitation in the wettest quarter (BIO16) for 13 species; mean temperature of the warmest quarter (BIO10) and precipitation in the warmest quarter (BIO18) for 12 species; and precipitation seasonality (BIO15) for 10 species, isothermality (BIO3; BIO/BIO7X100) for eight species, mean temperature of the wettest quarter (BIO8), mean diurnal range (BIO2), precipitation in the driest month (BIO14), precipitation in the wettest month (BIO13), annual mean temperature (BIO1), mean temperature of the driest quarter (BIO9), temperature seasonality (BIO4), and annual precipitation (BIO12) were the most informative bioclimatic variables, with correlations for >8, up to 14, species (Table 4). The model performance estimation for GAM, GLM, and MAXENT is presented in Table 5. According to both the AUC and TSS performance estimators, maximum entropy (MAXENT) is the best estimator for all species other than *D. brevicollis*, for which GAM is the best model. It should be noted that the performance values of both AUC and TSS were moderate, suggesting that these estimations need to be interpreted with caution.

## 4. Discussion and Conclusions

### 4.1. Faunal Composition of Anatolian Caelifera

Anatolia, by its geographic area size, constitutes roughly 0.001% of the world’s terrestrial area. However, with a total of 284 species/subspecies, Anatolia harbours 2.2% of the world’s Caelifera diversity, a proportion approximately 2000 times its geographic size. These contradictory proportions of geographic size and species percentages confirm that Anatolia constitutes a biodiversity hotspot for both Caelifera and Orthoptera. Anatolian Caelifera diversity comprises two families, Acrididae and Pamphagidae, representing 61.6% and 32%, respectively (Table 1, Figure 2). Taxa belonging to the remaining four families constitute only 6.4% of Anatolian Caelifera diversity. Although the species number of Acrididae occurring in Anatolia is higher than that of Pamphagidae, the former constitutes 2.5% of the world’s diversity, with 175 species, while the latter represents 14.4% of the world’s diversity, with 91 species (see [48] for species/subspecies diversity of the families). These percentages indicate that Anatolia constitutes an important fragment of the range of Acrididae and Pamphagidae, especially the latter, while serving as a peripheral range area for Tridactylidae, Tetrigidae, Derycoriythidae, and Pyrgomorphidae.

The above proportions per family indicate the range extension of families, but they do not provide insights into the evolution of these lineages in the area. Roughly 40% of Anatolian Caelifera are endemic, having evolved in this region. The proportion of endemic species or generic lineages carries important implications. Species poor families such as Tridactylidae, Tetrigidae, Derycoriythidae, and Pyrgomorphidae have no endemic representatives in Anatolia, leading us to consider Anatolia to be the marginal range area for these lineages. In contrast to these families, Pamphagidae and Acrididae boast a considerable number/proportion of endemic species. Pamphagidae occupies the top spot for endemism, with 71% percentage of endemic species, suggesting that Anatolia constitutes a centre of origin for this lineage. The presence of several endemic or predominantly Anatolian genera, namely, *Ebnerodes*, *Glyphothmethis*, *Paranocarodes*, *Paranothrotes*, *Pseudonothrotes*, *Nocarodes*, *Nocaracris*, and *Prionosthenus*, supports this claim. More importantly, the main species diversity of the family in the Palearctic occurs in Anatolia, indicating an autochthonous radiation on the margin of the Gondwanian region [21,31,32,71], especially as the main proportion of the diversity of this lineage is in Africa. The rate of endemic species belonging to Acrididae is lower (27%) but still considerable. There are no polytypic generic lineages endemic to Anatolia [72,73]. Instead, genera represented by several species in Anatolia, especially those belonging to Gomphocerinae, are widespread in the Palearctic. The species numbers of some of these genera, such *Chorthippus*, *Stenobothrus*, *Sphingonotus*, and *Omocestus*, are considerably high in Anatolia, and some of them are endemic, indicating that Anatolia is an important part of their diversity centre. Finally, there are no endemic species of the other four families in Anatolia.

### 4.2. Ecobiogeographic Classification of Anatolian Caelifera

Species diversity and composition in each ecobiogeographic fragment in Anatolia may differ due to several reasons. Two crucial factors are likely the area size and vegetation type of the fragment, considering that Caelifera members are herbivorous, and some are oligophagous, showing a preference for a limited number of certain plants. In terms of area size, Irano-Anatolia is the largest, followed by the Mediterranean, Euro-Siberia, and Mesopotamia. Vegetation type may be another factor determining Caelifera species diversity and composition, and for this reason we followed the ecobiogeographic classification was based on the phytogeographic classification developed by Zohary [30]. Fragments with steppe vegetation or predominantly steppe vegetation are expected to have greater Caelifera diversity. The Irano-Anatolian phytogeographic province is characterized by steppe vegetation, which also occurs in the southern parts of Euro-Siberia and the highlands of the Mediterranean. Mesopotamia is adjacent to the desert of the Arabian Peninsula. Consistent with the area sizes and vegetation types, Irano-Anatolia harbours the highest diversity, while Euro-Siberia corresponds to the second highest, the Mediterranean ranks third, and Mesopotamia has the poorest diversity. Although the Mediterranean is larger than Euro-Siberia in terms of area size, the species number is higher in the latter, possibly due to its vegetation composition, which provides habitats for cold-preferring species of Gomphocerinae. It should be noted that several species occur in more than one fragment, especially along the fragments’ adjacent areas. In all four geographic fragments, Acrididae and Pamphagidae are the dominant families, as is the case for the entirety of Anatolia.

The redefinition of the faunal structure of Anatolia necessitates a comparative evaluation across geographic fragments. Unlike Caelifera, Tettigoniinae was reported to be most diverse in the Mediterranean province, followed by Irano-Anatolia, Euro-Siberia, and Mesopotamia [25]. This result aligns with Uvarov’s [7] findings, indicating that the Mediterranean is more diverse compared to other provinces. A potential reason for this difference could be the limited presence of steppe vegetation in the Mediterranean province, which is crucial for Caelifera but less so for Ensifera. Ensifera includes several predatory species such as members of Pholidopterini and Drymadusini with a high number of endemic species [27,74]. In Anatolia, the proportion of endemic species belonging to Ensifera is approximately 80%, roughly twice that of Caelifera (approximately 40%, according to this study). This suggests that several ensiferan lineages originated in and radiated into Anatolia and are either Anatolian or predominantly Anatolian in their present distribution. Contrary to other families, the Pamphagidae lineage within Caelifera exhibits a diversity pattern similar to that of Ensifera. Approximately 20% of the world’s total pamphagid diversity occurs in Anatolia, and crucially, around 71% of them are endemic. Additionally, there are several genera of the family restricted to Anatolia or with only a few representatives in adjoining areas, indicating that Anatolia is an origin and radiation centre for this lineage.

Uvarov [7] classified Palearctic orthopteroid insects into four ecological categories: Boreal, Mediterranean, Steppe, and Eremian. In Figure 1A, the Boreal category corresponds to Euro-Siberia, and the Mediterranean corresponds to the respective region, but there are differences in our classification for the other two categories (Figure 2). Here, we have restricted Eremian to the lowlands of Mesopotamia, an area characterized by *Artemisia*-dominated dry habitats adjacent to the Arabian Peninsula desert [30].

The Irano-Anatolia region defined herein mainly includes Uvarov’s steppe region and part of the Eremian. Uvarov [7] uses the term “steppe subregion” for the Siberian habitat type. However, species occurring in Anatolia or predominantly in the Irano-Anatolian distribution, such as members of *Stenobothrus*, *Chorthippus*, and many other sublineages of Gomphocerinae, rarely extend beyond the Caucasus Mountains in the northeast or the highlands of the Balkans in the northwest [72,73]. Many Anatolian species have sister species in the adjoining Balkans, Caucasia, and other parts of the Black Sea Basin. Although many of them are not endemic, their ranges are limited to Anatolia and the surrounding areas, exhibiting characteristics of a gliding fauna, as stated by Kosswig [34].

Along with the endemic species in the area, the steppe fauna represents a regionally evolved diversity. Thus, we believe that these are resident lifeforms of the area, not that they evolved somewhere else (e.g., the Siberian steppes) and then arrived here, as suggested in earlier studies [20,21]. Additionally, Anatolia is possibly the centre of origin for many of them, either as species or multispecies lineages; see [33,35,54] for some examples.

In conclusion, Anatolia harbours Caelifera fauna, especially those occurring in the Mediterranean and Irano-Anatolia, mainly originating in the area. While there are some members that arrived from Africa (mainly North Africa) and Central/East Asia, as mentioned in earlier biogeographic studies [7,20,21], they constitute an insignificant fraction. Another issue related to the steppe elements in Anatolia is the definition of internal refugia by Weidner [20]. These refugia mainly correspond to some altitudinal chains with steppe vegetation in the Mediterranean and Euro-Siberian regions, remaining outside of Irano-Anatolia (see Figure 1B). These mountain chains are characterized by the existence of cold-demanding members of Gomphocerinae, either as endemics or as fragmented populations of some widespread species, which were defined as taxa with a boreo-alpine distribution by De Latin [75]. Thus, we think these refugia do not represent different faunal characteristics that should be evaluated separately. Contrary to other eco-geographic regions, there are no species endemic to Mesopotamia, and the species occurring in the area are common in large parts of the Palearctic.

### 4.3. Pest and Pullulating Species of Caelifera in Anatolia 

Although publications on pest orthopterans in Anatolia date back to the time of the First World War [36,37,38,41,42,43,44,45,46], these studies generally focused on classical swarming locust species, such as *Dociostaurus maroccanus*, *Calliptamus italicus*, and *Schistocerca gregaria*. As locally proliferating species were rarely examined or reported (as reviewed in [40]), our results can provide significant indications for pest management organisations. Personal observations made by the first author over the course of 35 years throughout Anatolia revealed that several species have the ability to become pests. This is the reason why the Directorate of the Plant Protection Central Research Institute applied insecticides to proliferating grasshopper populations in various locations in Anatolia between 2013 and 2020 (see Figure 6 in [40]). According to experts from the Directorate of the Plant Protection Central Research Institute [76], insecticide application was rarely employed for certain species, especially *Locusta migratoria*, and instead was used for multispecies grasshopper communities that locally became abundant. Data provided by experts from the Directorate of the Plant Protection Central Research Institute and personal observations made by the first author indicate that these grasshopper communities mainly consist of *Callipttamus* spp., *Oedipoda* spp., *N. anatolicus*, and *H. pterosthica* in lowland plains and *D. brevicollis*, *Chortippus* spp., *E. pulvinatus*, and *A. labiata* in highland areas. The pest state is observed during the summer, especially after the wild vegetation has dried out, and these animals gather in watered green agricultural areas. Rather than damaging lowland agricultural areas, species proliferating in highlands harm the pastures in the countryside.

In this study, we attempted to predict the future pest potential of 14 grasshopper species in Anatolia by modelling their distributions for both the present and future (2070). Of the 14 species modelled (see Figure 5), *Calliptamus italicus*, *C. barbarus*, *C. tenuicercis*, *Notostaurus anatolicus*, *Oedipoda miniata*, and *O.schochii* occur in lowland (<1200 m) arid habitats around agricultural areas; *Arcyptera labiata*, *Dociostaurus maroccanus*, and *D. brevicollis* occur in semi-arid areas with step vegetation at moderate altitudes; *Heteracris pterosticha* and *Locusta migratoria* occur in watered humid plains at moderate/lowland altitudes; and *Chorthippus dichrous*, *Ch. karelini*, and *Euchorthippus pulvinatus* occur in montane meadows at highland. The modelling results showed varied predictions for each species, including insignificant changes for three, reductions for five, and enlargements for the remaining five. It is important to note that these predictions come with certain limitations, such as relatively low statistical support (in this case, with respect to AUC and TTS), possibly due to the limited number of locality records and the absence of certain ecological factors in the analyses. The oligophage feeding preference of the acridid species, which was not explicitly considered in the modelling, might have influenced the accuracy of predictions. Additionally, other ecological factors like competition and predators, which were not accounted for in the conventional analyses, could impact species occurrence. 

Despite these limitations, this study suggests important clues in the modelling results. The reduction in the distribution size of certain species associated with montane meadows, such as *Chorthippus dichrous*, *Ch. karelini*, and *Dociostaurus brevicollis*, aligns with expectations considering the potential effects of global warming on such habitats. On the other hand, the enlargement in distribution size for species like *Dociostaurus maroccanus*, *Locusta migratoria*, *Heteracris pterosticha*, *Oedipoda miniata*, *O. schochii*, and *Euchorthippus pulvinatus* was expected due to their wide ecological tolerance. A recent proliferation of *H. pterosticha* [40] supports this estimation. Of these five species, though *E. pulvinatus* occurs in highlands, it prefers relatively arid areas compared to two species of *Chorthippus* preferring moist meadows. *D. marrocanus* and *L. migratoria* are already-known pest species and continuously under management by the Directorate of the Plant Protection Central Research Institute [40,76]. This study emphasizes the need for caution in interpreting these predictions and recommends monitoring the population densities of certain species, including *D. maroccanus*, *L. migratoria*, *H. pterosticha*, *C. italicus*, *N. anatolicus*, and *Oedipoda* spp., to determine their pest potential in the future. The authors acknowledge that more comprehensive analyses, incorporating extensive occurrence data and additional ecological factors, are essential to acquire a better understanding of the biogeography and pest potential of short-horned grasshoppers.

## Figures and Tables

**Figure 1 insects-15-00055-f001:**
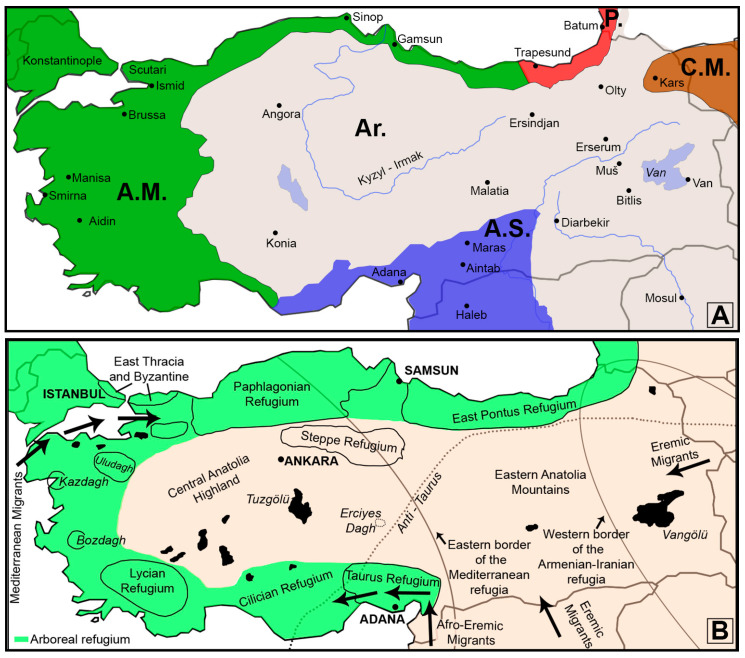
Eco-biogeographic classification of Anatolian short-horned grasshoppers modified and re-drawn according to (**A**) Uvarov [7] (A.M. (dark green)—Anatolia-Mediterranean; A.S. (blue)—Syrian-Anatolia; Ar. (grey)—Armenian district; C.M. (brown)—District of Caucasus Minor, P. (red)—Pontian district) and (**B**) Weidner [20] (green: Arboreal refugium, nude: Central and Eastern Anatolia, Caucasus and Middle East).

**Figure 2 insects-15-00055-f002:**
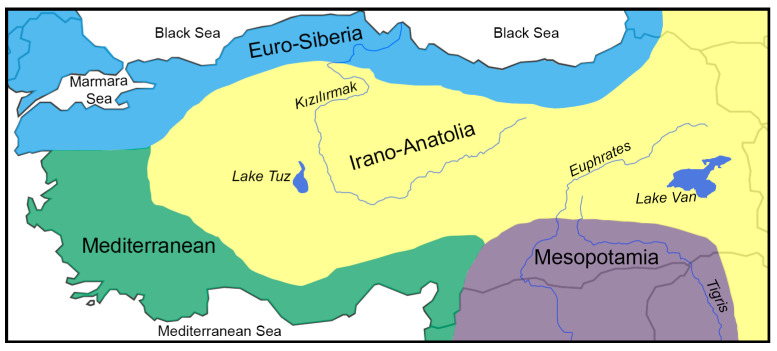
The eco-biogeographic sections used in this study to define habitat preferences of Anatolian short-horned grasshoppers (the sections are defined according to phytogeographical provinces in Anatolia by considering the work by Zohary [30], Çıplak [26], and Kaya & Raynal [49].

**Figure 3 insects-15-00055-f003:**
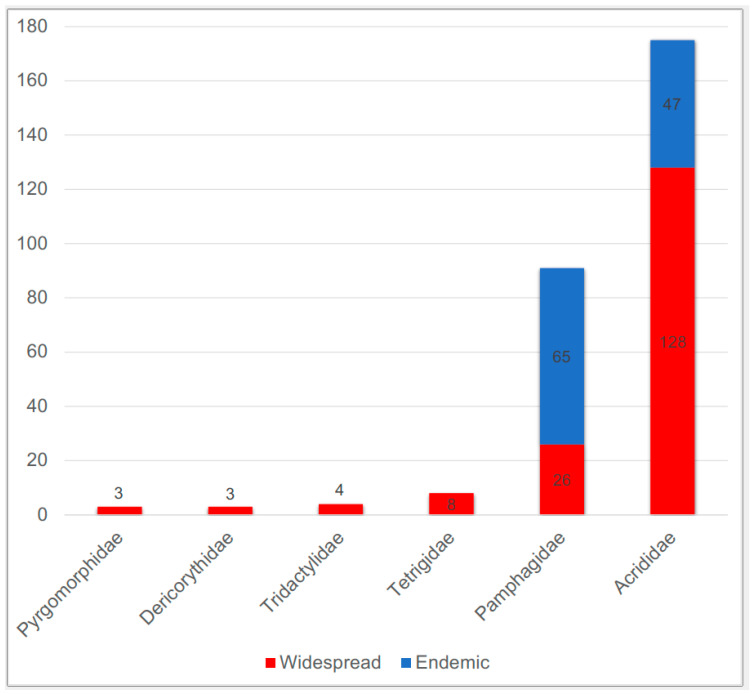
The number of widespread, endemic, and total species per caeliferan families occurring in Anatolia (for details, see Appendix B Table A1).

**Figure 4 insects-15-00055-f004:**
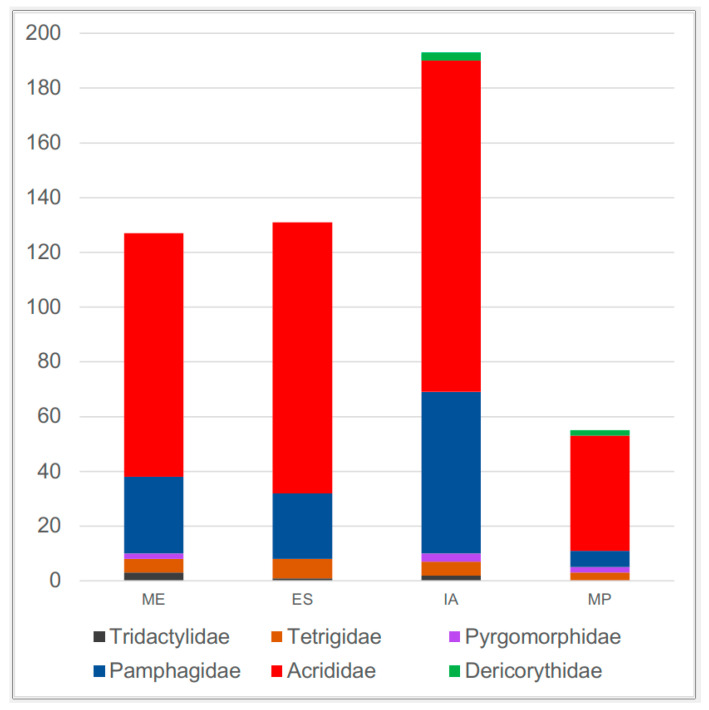
The number of species per Caelifera family in each of the four phytogeographical provinces of Anatolia (ME: Mediterranean, ES: Euro-Siberia, MP: Mesopotamia, and IA: Irano-Anatolia) (for details, see Appendix B Table A1 and Figure 2).

**Figure 5 insects-15-00055-f005:**
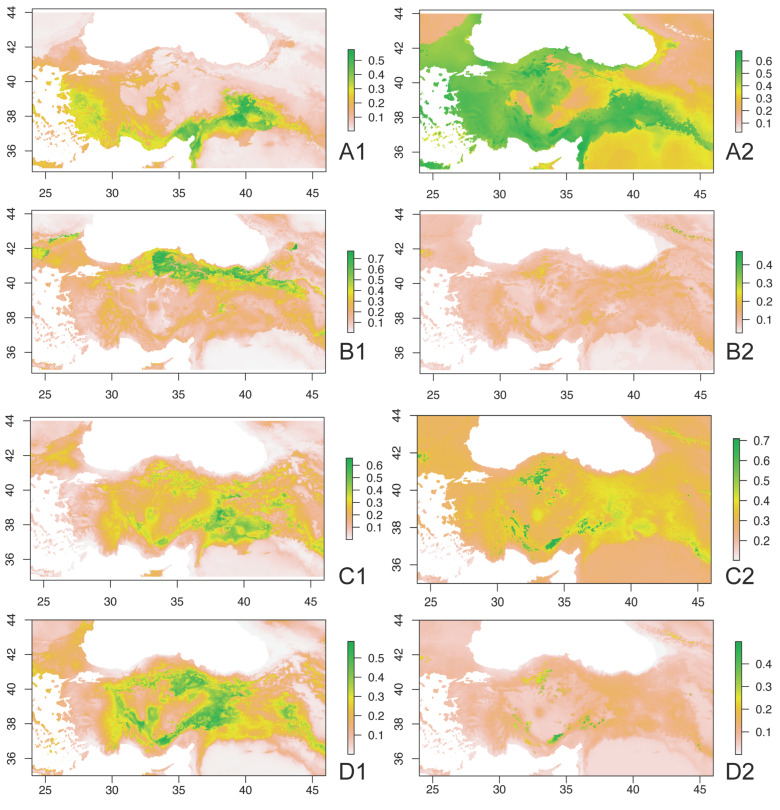
Present (**1**) and future (2070) (**2**) distribution predictions for 14 species of Acrididae with pest potential. (**A**)—*Locusta migratoria*, (**B**)—*Calliptamus italicus*, (**C**)—*C. barbarus*, (**D**)—*C. tenuicercis.* (**E**)—*Dociostaurus maroccanus*, (**F**)—*D. brevicollis*, (**G**)—*Notostaurus anatolicus*, (**H**)—*Heteracris pterosticha*, (**I**)—*Arcyptera labiata*. (**J**)*—Oedipoda miniata*, (**K**)—*Oedipoda schochii*, (**L**)—*Chorthippus dichrous*, (**M**)—*Ch. karelini*, and (**N**)—*Euchorthippus pulvinatus*.

**Table 1 insects-15-00055-t001:** The number of widespread and endemic species per Caelifera family occurring in Anatolia.

Family	N, Widespread	N, Endemic	Total
Tridactylidae	4	0	4
Pyrgomorphidae	3	0	3
Dericorythidae	3	0	3
Tetrigidae	8	0	8
Pamphagidae	26	65	91
Acrididae	128	47	175
Total	172	112	284

**Table 2 insects-15-00055-t002:** The number of species per family in each of four phytogeographical regions of Anatolia.

	Irano-Anatolia	Euro–Siberia	Mediterranean	Mesopotamia
Dericorythidae	3	-	-	2
Pyrgomorphidae	3	-	2	2
Tridactylidae	2	1	3	-
Tetrigidae	5	7	5	3
Pamphagidae	59	24	28	6
Acrididae	121	99	89	42
Total	193	131	127	55

**Table 3 insects-15-00055-t003:** The description of 19 bioclimatic variables from the WorldClim database and those used in the species distribution modelling.

Bioclimatic Variables	Description	Bioclimatic Variables	Description
BIO1	Annual mean temperature	BIO11	Mean temperature of coldest quarter
BIO2	Mean diurnal range (mean of monthly (max temp–min temp))	BIO12	Annual precipitation
BIO3	Isothermality (BIO2/BIO7) (×100)	BIO13	Precipitation in wettest month
BIO4	Temperature seasonality (standard deviation ×100)	BIO14	Precipitation in driest month
BIO5	Max temperature of warmest month	BIO15	Precipitation seasonality (coefficient of variation)
BIO6	Min temperature of coldest month	BIO16	Precipitation in wettest quarter
BIO7	Temperature annual range (BIO5-BIO6)	BIO17	Precipitation in driest quarter
BIO8	Mean temperature of wettest quarter	BIO18	Precipitation in warmest quarter
BIO9	Mean temperature of driest quarter	BIO19	Precipitation in coldest quarter
BIO10	Mean temperature of warmest quarter		

**Table 4 insects-15-00055-t004:** The uncorrelated bioclimatic factors and their variance inflation factor (VIF) scores used for each species’ modelling (* the correlated factor for respective species; LM—*Locusta migratoria*, CI—*Calliptamus italicus*, CB—*C. barbarus*, CT—*C. tenuicercis*, DM—*Dociostaurus maroccanus*, DB—*D. brevicollis*, HP—*Heteracris pterosticha*, NA—*Notostaurus anatolicus*, AL—*Arcyptera labiata*, OM*—Oedipoda miniata* OS—*O*. *schochii*, ChD—*Chorthippus dichrous*, ChK—*Ch. karelini*, and EP—*Euchorthippus pulvinatus*).

Bioclimatic Variables	LM	CI	CB	CT	DM	DB	HP	NA	AL	OM	OS	ChD	ChK	EP
BIO1	4.405	3.614	2.061	1.487	*	2.448	*	2.562	2.986	*	4.217	2.828	*	*
BIO2	2.069	1.758	1.959	2.132	2.917	1.588	1.178	1.566	2.514	*	*	2.011	2.112	2.135
BIO3	1.840	1.380	*	*	*	*	*	*	*	2.798	2.629	*	1.996	1.301
BIO4	*	*	1.382	1.508	2.682	1.920	4.716	1.684	2.791	3.056	*	1.856	*	*
BIO5	*	*	*	*	*	*	*	*	*	*	*	*	*	*
BIO6	*	*	*	*	*	*	*	*	*	*	*	*	*	*
BIO7	*	*	*	*	*	*	*	*	*	*	1.793	*	*	*
BIO8	2.510	1.308	2.236	2.392	3.224	3.162	1.326	2.115	4.324	1.365	4.000	1.898	2.439	2.527
BIO9	*	4.189	*	*	1.677	2.341	1.908	2.171	*	3.072	2.975	2.872	2.814	4.967
BIO10	*	*	*	*	*	*	*	*	*	*	*	*	2.826	3.171
BIO11	*	*	*	*	*	*	*	*	*	*	*	*	*	*
BIO12	2.133	*	2.061	7.391	*	*	*	8.389	4.256	3.768	5.918	7.237	4.313	*
BIO13	*	1.579	6.978	5.071	*	3.363	4.929	6.112	5.129	*	4.947	4.748	*	4.331
BIO14	5.790	2.615	4.331	2.555	*	3.115	1.905	2.873	4.391	6.002	2.954	3.129	9.876	
BIO15	6.361	*	*	*	*	*	*	*	*	9.466	*	*	5.524	5.768
BIO16	*	*	*	*	2.981	*	*	*	*	*	*	*	*	*
BIO17	*	*	*	*	*	*	*	*	*	*	*	*	*	*
BIO18	*	*	*	*	4.062	*	*	*	*	*	*	*	*	7.324
BIO19	*	*	*	*	*	*	*	*	*	*	*	*	*	*

**Table 5 insects-15-00055-t005:** The model performance parameters (area-under-the-curve (AUC) and true skill statistic (TSS) values) for each of the Generalized Additive Models (GAMs), Generalized Linear Models (GLMs), and Maximum Entropy (MaxEnt) estimated for each species.

Species	Methods	AUC	TSS	Species	Methods	AUC	TSS
*Arcyptera labiata*	GLM	0.79	0.61	*Dociostaurus maroccanus*	GLM	0.65	0.49
	GAM	0.84	0.71		GAM	0.65	0.46
	MAXENT	0.87	0.71		MAXENT	0.77	0.65
*Calliptamus barbarus*	GLM	0.69	0.41	*Euchortippus pulvinatus*	GLM	0.78	0.62
	GAM	0.79	0.54		GAM	0.79	0.64
	MAXENT	0.8	0.54		MAXENT	0.86	0.72
*Calliptamus italicus*	GLM	0.74	0.45	*Heteracris pterosticha*	GLM	0.75	0.61
	GAM	0.84	0.61		GAM	0.75	0.58
	MAXENT	0.82	0.56		MAXENT	0.82	0.71
*Calliptamus tenuicercis*	GLM	0.71	0.47	*Oedipoda schochii*	GLM	0.79	0.63
	GAM	0.79	0.55		GAM	0.84	0.69
	MAXENT	0.81	0.59		MAXENT	0.85	0.72
*Chorthippus dichrous*	GLM	0.74	0.45	*Oedipoda miniata*	GLM	0.69	0.41
	GAM	0.79	0.52		GAM	0.76	0.52
	MAXENT	0.8	0.54		MAXENT	0.75	1.2
*Chorthippus karelini*	GLM	0.83	0.65	*Locusta migratoria*	GLM	0.78	0.58
	GAM	0.84	0.66		GAM	0.79	0.6
	MAXENT	0.84	0.68		MAXENT	0.83	0.67
*Dociostaurus brevicollis*	GLM	0.78	0.56	*Notostaurus anatolicus*	GLM	0.71	0.47
	GAM	0.86	0.69		GAM	0.78	0.56
	MAXENT	0.85	0.66		MAXENT	0.8	0.58

## Data Availability

All data used in producing this paper are presented in the paper.

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
