# Peer review of "Anatolian Short-Horned Grasshoppers Unveiled: Integrating Biogeography and Pest Potential"

_insects, 2024, doi:10.3390/insects15010055_

Round 1

Reviewer 1 Report

Comments and Suggestions for Authors

Overall, I found this paper to be informative and I think it will be useful for at least the Orthoptera community, so I recommend it be published, albeit with very minor revisions. I highly recommend the authors review and incorporate the 43 comments I made on the PDF to make it even stronger. In particular, I think they need to make it very clear how many pest and pest potential species they're focusing on - I counted 12, 13, or 14 depending on which part of the paper I was in, which is very confusing. The majority of the comments are focused on grammatical issues and aesthetics. I also noted that in at least one case, the in-text citation was incorrect, which may mean others are also referenced incorrectly, so they should be double-checked.

Comments on the Quality of English Language

I was highly impressed with the quality of the English Language found in this manuscript. I've found more issues in the manuscripts of native English speakers, so keep up the great work!

Author Response

Thank you very much for the constructive comments. All were considered carefully and the manuscript was revised accordingly.

Reviewer 2 Report

Comments and Suggestions for Authors

General comments

The study by Ciplak & Uluar is a valuable contribution to the understanding of biogeographical patterns of Anatolian Orthoptera, offering an applied perspective to the topic by estimating future distribution of the species with pest potential. I recommend accepting the paper for publication in Insects, after some necessary improvements are made. Objectives of the research are well-defined, statistical analyses are sound and clearly explained. While generally well-written, the introduction is too comprehensive and unnecessarily detailed in some respects (e.g. historical overview of Orthoptera research in Anatolia), and some information provided in the discussion would more appropriately fit in the introduction (e.g. the background for pest potential research) or the results (for instance, simply listing the most important findings with exact numbers and percentages should be avoided in the discussion). Reading the discussion in its current form, it is hard to pinpoint the most important findings of your research, since all the findings are given equal or similar weight. However, you should identify your key findings and then rewrite certain sections of the discussion to focus on these findings, discussing them in the context of existing biogeographical classifications, known biogeographical patterns for other animal groups, existing studies evaluating future pest potential of Orthoptera in other regions etc. Also, the ordering of the tables and figures could be improved. Please find more detailed comments below.

Abstract

Line 40: It should be Calliptamus italicus instead of Calipptamus italicus (two “l” instead of two “p”). Please check the spelling of species names throughout the manuscript.

Introduction

Line 46: I recommend to start with a more general paragraph on Anatolian Orthoptera before moving on to the issue of biogeography (for instance, start with the second paragraph, then move on to the first; see the comments below).

Lines 55-56: In the previous sentence you mention “sublineages” of Orthoptera, and here you write about “lineages belonging to Orthoptera”. Please use the terms “lineage” and “sublineage” consistently so that the reader is not confused about the taxonomic level you are referring to.

Lines 62-83: Please shorten this paragraph, leaving only the basic information about the trends in Orthoptera research in Anatolia and the current state of knowledge – in my opinion, such a comprehensive historical overview is not necessary in the context of the current research.

Lines 66-67: There is no need to repeat the information already given earlier in the text (“…as mentioned above, constitutes a hallmark in respect to eco-biogeographic analysis of orthopterous insects…”). Rather think about revising the order of the paragraphs, so that you start with this paragraph (a more general paragraph about Orthoptera research in Anatolia), and then move on to the issue of biogeography, which is the main topic of your paper (see the comment above).

Line 135: The transition from biogeography to swarming is very abrupt, you simply introduce the issue of swarming with a single, rather general and vague sentence: “Some of the swarming Caeliferan locust occurs in Anatolia.”. Please make this transition smoother, preferably by associating the importance of biogeographical knowledge for pest management in general, then again moving on Orthoptera and the swarming.

Line 156: What do you consider a “lineage” in this context? Do you refer to taxonomic ranks above or below the species level, or both? Namely, species are also lineages included within a genus as a broader lineage, which is again a lineage within a family etc. On the other hand, a species can also comprise several lineages that can be regarded semi-species, subspecies etc. You should define terms clearly in the introduction and then use them in an unambiguous manner throughout the manuscript.

Results

Lines 222-223: Please consider moving Table 1 to the supplementary material, simply due to its size – at the moment it takes up as much as 11 pages of the manuscript. It is important to provide the summary data (e.g. the number of species per phytogeographical region etc.; currently Figure 3) in the main text, but in my opinion, table listing all the species with presence/absence for each region is more suited to the supplement, where readers can take a look at it if interested in any particular species. In this case, Table 2 and Table 3 should then be changed to Table 1 and Table 2, respectively (and so on for other tables, accordingly).

Table 1: There is a technical problem with the table, it seems that the numbers designating the species have overlapped with the numbers representing the lines in the document, making it very difficult to follow the order of the species. The same applies for Tables 2-6. Please correct this. Also, the table is rather narrow, and the species with longer names currently take up two rows instead of one. I suggest to avoid this by making the table broader.

Lines 269-270: I have noticed that you use the abbreviation MP for Mesopotamia in the table caption, but MS is written in the table. Please make this consistent throughout the manuscript.

Line 298 (and elsewhere if applicable): Figure and table captions should be easy to understand on their own, without having to look something up in the main text or in the previous page. Therefore, please write the full caption whenever the table is continued in the new page. This caption should be followed by “Continued from the previous page.”

Lines 259-256: Please consider using the full names of the variables (mean temperature, temperature seasonality, annual temperature range etc.) in this paragraph instead of the codes (BIO1, BIO4, BIO7 etc.), because the latter are not informative to the reader. I am aware that the codes are explained in Table 4, but having to look up each code in the table negatively affects the readability of the text.

Line 563: Table caption mentions species, but the reader has to read the main text to find out that you refer to species belonging to Orthoptera, or more precisely, Caelifera. Figure and table captions should be easy to understand on their own, without having to look something up in the main text or in the previous page (see the comment above). Please write “caeliferan species” or “species of Caelifera”, to specify which taxon these species belong to. Apply to other figure/table captions throughout the manuscript.

Figure 3 and 4: Why are these figures, which are referred to at the very beginning of the results (in the first two paragraphs), placed after the tables showing bioclimatic variable scores and model performance parameters (which are referred to in the last paragraph of the results)? Figures and tables should closely follow the information provided in the text, otherwise it is hard to keep track of the main findings. Please re-arrange the tables and figures so that the respective table/figure appears after the paragraph in which it is mentioned for the first time.

Discussion

Line 730: What are you referring to as “the above rates”? Please specify or rephrase in a clearer way.

Lines 755-757: The information on how you conducted a certain aspect of your research (in this case, ecobiographic classification) should be provided in the Material and methods, not in the Discussion. Please move this to the Material and methods or, if this information is already provided there, please remove it, since such repetitions should be avoided.

Lines 716-718, 761-768: You should avoid listing the results in the Discussion, especially providing the exact number of species or percentage for each group (“… Irano-Anatolia harbours the highest diversity with 193 species, Euro-Siberia is the second with 131…”), since this should have been given in the Results section of the paper. In the Discussion, you should put these findings in a biogeographical context, discussing your findings in the light of existing biogeographical classifications, comparing them with biogeographical patterns for other insect (or indeed, animal) groups etc. You can perhaps emphasize the most important numbers or proportions, but only to illustrate your points, not to give an overview of the results (once again, this belongs to the Results).

Line 798: Figure and table captions should be easy to understand on their own, without having to look something up in the main text or in the previous page. Therefore, please write the full caption whenever the figure is continued in the new page. This caption should be followed by “Continued from the previous page.”

Figure 5: Why is this figure included in the Discussion, when it should be shown already in the Results, more precisely after the final paragraph (Lines 553-561)?

Lines 812-813: What do you base this assumption (that these are resident forms) on? Please elaborate.

Lines 829-837: This entire section elaborates the background for pest potential research on Anatolian Orthoptera, and should therefore be moved to the Introduction. In the Discussion you should focus on the findings of the current study, discussing them in the context of previous research, not only in Anatolia but also in other regions, going from specific and local towards more general and global.

Lines 847-849: This text explains the background for the analysis, and should therefore be moved to the Material and methods (see the comments above).

Comments on the Quality of English Language

Although the manuscript is generally written in competent and clear English, the wording can be improved in some instances (please see more detailed comments below). Therefore, I suggest proof-reading the manuscript once again to further improve English.

Line 122: Are you sure that “frustrates” is an appropriate word choice here? Maybe “to hinder” or “to make difficult” would be more appropriate in this context? Although the manuscript is generally written in competent and clear English, the wording can be improved in some instances. Therefore, I suggest proof-reading the manuscript once again to further improve English.

Line 127: Please add “that” between “lineages” and “originated”.

Line 128: I have some trouble understanding certain parts of this sentence. Perhaps it is simply a matter of wording, but your sentence seems to imply that “in many lineages of Ensifera” it is somehow “observed” that “some lineages of Gomphocerinae” have evolved in Anatolia. Please rephrase to make your meaning clearer.

Line 131: To make your case stronger, maybe it would be worthwhile to rephrase: “…as suggested by both early (REF) and recent studies (REF).”

Line 732: “Rate” typically designates a measure put in relation with another measure. In this case, I believe you are talking about a proportion (40%) of endemic species? Please correct throughout the text.

Line 742: I believe you should again use “proportion” instead of “rate” here (please see the previous comment). Please correct throughout the text.

Author Response

Thank you very much for the constructive comments. All were considered carefylly and the manuscript was revised accordingly. Details are below!

The study by Ciplak & Uluar is a valuable contribution to the understanding of biogeographical patterns of Anatolian Orthoptera, offering an applied perspective to the topic by estimating future distribution of the species with pest potential. I recommend accepting the paper for publication in Insects, after some necessary improvements are made. Objectives of the research are well-defined, statistical analyses are sound and clearly explained. While generally well-written, the introduction is too comprehensive and unnecessarily detailed in some respects (e.g. historical overview of Orthoptera research in Anatolia), and some information provided in the discussion would more appropriately fit in the introduction (e.g. the background for pest potential research) or the results (for instance, simply listing the most important findings with exact numbers and percentages should be avoided in the discussion). Reading the discussion in its current form, it is hard to pinpoint the most important findings of your research, since all the findings are given equal or similar weight. However, you should identify your key findings and then rewrite certain sections of the discussion to focus on these findings, discussing them in the context of existing biogeographical classifications, known biogeographical patterns for other animal groups, existing studies evaluating future pest potential of Orthoptera in other regions etc. Also, the ordering of the tables and figures could be improved. Please find more detailed comments below.

BC- Thanks for supportive comments!

Abstract

Line 40: It should be Calliptamus italicus instead of Calipptamus italicus (two “l” instead of two “p”). Please check the spelling of species names throughout the manuscript.

BC- Thanks, corrected!

Introduction

Line 46: I recommend to start with a more general paragraph on Anatolian Orthoptera before moving on to the issue of biogeography (for instance, start with the second paragraph, then move on to the first; see the comments below).

BC- Thanks! Revised accordingly!

Lines 55-56: In the previous sentence you mention “sublineages” of Orthoptera, and here you write about “lineages belonging to Orthoptera”. Please use the terms “lineage” and “sublineage” consistently so that the reader is not confused about the taxonomic level you are referring to.

BC- Thanks! The order of first and second paragraph changed!

Lines 62-83: Please shorten this paragraph, leaving only the basic information about the trends in Orthoptera research in Anatolia and the current state of knowledge – in my opinion, such a comprehensive historical overview is not necessary in the context of the current research.

BC- Thanks! The order of first and second paragraph changed!

Lines 66-67: There is no need to repeat the information already given earlier in the text (“…as mentioned above, constitutes a hallmark in respect to eco-biogeographic analysis of orthopterous insects…”). Rather think about revising the order of the paragraphs, so that you start with this paragraph (a more general paragraph about Orthoptera research in Anatolia), and then move on to the issue of biogeography, which is the main topic of your paper (see the comment above).

BC- Thanks! Text revised accordingly!

Line 135: The transition from biogeography to swarming is very abrupt, you simply introduce the issue of swarming with a single, rather general and vague sentence: “Some of the swarming Caeliferan locust occurs in Anatolia.”. Please make this transition smoother, preferably by associating the importance of biogeographical knowledge for pest management in general, then again moving on Orthoptera and the swarming.

BC- Thanks! Revised by reordering and rephrasing the paragraphs!

Line 156: What do you consider a “lineage” in this context? Do you refer to taxonomic ranks above or below the species level, or both? Namely, species are also lineages included within a genus as a broader lineage, which is again a lineage within a family etc. On the other hand, a species can also comprise several lineages that can be regarded semi-species, subspecies etc. You should define terms clearly in the introduction and then use them in an unambiguous manner throughout the manuscript.

BC- Thanks! Revised accordingly!

Results

Lines 222-223: Please consider moving Table 1 to the supplementary material, simply due to its size – at the moment it takes up as much as 11 pages of the manuscript. It is important to provide the summary data (e.g. the number of species per phytogeographical region etc.; currently Figure 3) in the main text, but in my opinion, table listing all the species with presence/absence for each region is more suited to the supplement, where readers can take a look at it if interested in any particular species. In this case, Table 2 and Table 3 should then be changed to Table 1 and Table 2, respectively (and so on for other tables, accordingly).

BC- Respectfully, we prefer to retain the table in the main text as it is basic for the paper. However, we transferred it to end of the whole text as an Appendix!

Table 1: There is a technical problem with the table, it seems that the numbers designating the species have overlapped with the numbers representing the lines in the document, making it very difficult to follow the order of the species. The same applies for Tables 2-6. Please correct this. Also, the table is rather narrow, and the species with longer names currently take up two rows instead of one. I suggest to avoid this by making the table broader.

BC- Thanks! It is because of the template file!  

Lines 269-270: I have noticed that you use the abbreviation MP for Mesopotamia in the table caption, but MS is written in the table. Please make this consistent throughout the manuscript.

BC- Thanks! Corrected!

Line 298 (and elsewhere if applicable): Figure and table captions should be easy to understand on their own, without having to look something up in the main text or in the previous page. Therefore, please write the full caption whenever the table is continued in the new page. This caption should be followed by “Continued from the previous page.”

BC- Thanks! Revised accordingly!

Lines 259-256: Please consider using the full names of the variables (mean temperature, temperature seasonality, annual temperature range etc.) in this paragraph instead of the codes (BIO1, BIO4, BIO7 etc.), because the latter are not informative to the reader. I am aware that the codes are explained in Table 4, but having to look up each code in the table negatively affects the readability of the text.

BC- Thanks! Revised accordingly!

Line 563: Table caption mentions species, but the reader has to read the main text to find out that you refer to species belonging to Orthoptera, or more precisely, Caelifera. Figure and table captions should be easy to understand on their own, without having to look something up in the main text or in the previous page (see the comment above). Please write “caeliferan species” or “species of Caelifera”, to specify which taxon these species belong to. Apply to other figure/table captions throughout the manuscript

BC- Thanks! Revised accordingly!

Figure 3 and 4: Why are these figures, which are referred to at the very beginning of the results (in the first two paragraphs), placed after the tables showing bioclimatic variable scores and model performance parameters (which are referred to in the last paragraph of the results)? Figures and tables should closely follow the information provided in the text, otherwise it is hard to keep track of the main findings. Please re-arrange the tables and figures so that the respective table/figure appears after the paragraph in which it is mentioned for the first time.

BC- Thanks! Revised accordingly!

Discussion

Line 730: What are you referring to as “the above rates”? Please specify or rephrase in a clearer way.

BC- Thanks! Revised for clarity!

Lines 755-757: The information on how you conducted a certain aspect of your research (in this case, ecobiographic classification) should be provided in the Material and methods, not in the Discussion. Please move this to the Material and methods or, if this information is already provided there, please remove it, since such repetitions should be avoided.

BC- Respectfully, we prefer to retain this sentences, as it is also a case supporting our conclusions.  

Lines 716-718, 761-768: You should avoid listing the results in the Discussion, especially providing the exact number of species or percentage for each group (“… Irano-Anatolia harbours the highest diversity with 193 species, Euro-Siberia is the second with 131…”), since this should have been given in the Results section of the paper. In the Discussion, you should put these findings in a biogeographical context, discussing your findings in the light of existing biogeographical classifications, comparing them with biogeographical patterns for other insect (or indeed, animal) groups etc. You can perhaps emphasize the most important numbers or proportions, but only to illustrate your points, not to give an overview of the results (once again, this belongs to the Results).

BC- Thanks! A full review of Anatolian biogeography is beyond aims of present study, thus we limited our discussion with Caelifera. However, the text revised to avoid Results-like sentences.  

Line 798: Figure and table captions should be easy to understand on their own, without having to look something up in the main text or in the previous page. Therefore, please write the full caption whenever the figure is continued in the new page. This caption should be followed by “Continued from the previous page.”

BC- Thanks! Revised accordingly!

Figure 5: Why is this figure included in the Discussion, when it should be shown already in the Results, more precisely after the final paragraph (Lines 553-561)?

BC- Thanks! Revised accordingly!

Lines 812-813: What do you base this assumption (that these are resident forms) on? Please elaborate.

BC- Thanks! Revised for clarity!

Lines 829-837: This entire section elaborates the background for pest potential research on Anatolian Orthoptera, and should therefore be moved to the Introduction. In the Discussion you should focus on the findings of the current study, discussing them in the context of previous research, not only in Anatolia but also in other regions, going from specific and local towards more general and global.

BC- Thanks! The text revised for clarity!

Lines 847-849: This text explains the background for the analysis, and should therefore be moved to the Material and methods (see the comments above).

BC- Thanks! The text revised for clarity!

Comments on the Quality of English Language

Although the manuscript is generally written in competent and clear English, the wording can be improved in some instances (please see more detailed comments below). Therefore, I suggest proof-reading the manuscript once again to further improve English.

Line 122: Are you sure that “frustrates” is an appropriate word choice here? Maybe “to hinder” or “to make difficult” would be more appropriate in this context? Although the manuscript is generally written in competent and clear English, the wording can be improved in some instances. Therefore, I suggest proof-reading the manuscript once again to further improve English.

BC- Thanks! Revised accordingly!

Line 127: Please add “that” between “lineages” and “originated”.

BC- Thanks! Revised accordingly!

Line 128: I have some trouble understanding certain parts of this sentence. Perhaps it is simply a matter of wording, but your sentence seems to imply that “in many lineages of Ensifera” it is somehow “observed” that “some lineages of Gomphocerinae” have evolved in Anatolia. Please rephrase to make your meaning clearer.

BC- Thanks! Revised accordingly!

Line 131: To make your case stronger, maybe it would be worthwhile to rephrase: “…as suggested by both early (REF) and recent studies (REF).”

BC- Thanks! Revised accordingly!

Line 732: “Rate” typically designates a measure put in relation with another measure. In this case, I believe you are talking about a proportion (40%) of endemic species? Please correct throughout the text.

BC- Thanks! Revised accordingly!

Line 742: I believe you should again use “proportion” instead of “rate” here (please see the previous comment). Please correct throughout the text.

BC- Thanks! Revised accordingly!

Reviewer 3 Report

Comments and Suggestions for Authors

 General comments

The paper by Ciplak and Uluar has three goals: update the list of Anatolian Caelifera, biogeographically classify Caelifera fauna, and predict range changes of economically injurious grasshoppers through ecological niche modeling. The impact of this work may be broadened by inclusion of general biogeography. The Introduction is entirely and narrowly focused on grasshoppers. This is a grasshopper specific work to be sure, but biogeographic provinces are best drawn from shared distributions of diverse life forms. Despite Orthoptera representing a basis for Anatolian biogeography, consider beginning the introdution with background about the biogeograhy of Anatolia in general as defined by the distributions of numerous organisms. The first time the reader sees the vegetation-based provinces that the authors use is in the Methods. Introducing this scheme in the Intro will set context for the historical and current Orthoptera studies, explain how the authors define biogeographical provinces, which is important given the dissimilarities among the provinces as defined by historical authors working on the Orthoptera.

It is worth mentioning and referencing the endemism of Orthoptera in other parts of the world and not just in Anatolia. For example, the Orthoptera of California, USA are highly endemic as reported by Harrison 2013 “Plant and Animal Endemism in California.” Orthoptera are generally useful in defining biogeography because they are primary consumers, thus following regional floras rather closely. Grasshoppers also codistribute with soil types, perhaps worth mentioning or considering when detailing biogeographical provinces. This strengthens the choice to work on Orthoptera by illustrating how useful they are for understanding biogeogrpahy worldwide.

In several places in the manuscript claims are given without evidence, particularly claims regarding origins or evolutionary history. In some cases reference(s) may simply be missing, in others the data may not exist, as presented here the reader cannot evaluate.

I caution the authors to carefully consider alternative biogeographic hypotheses and be clear with their definitions. A lineage may be diverse in a region without that region being its center of origin. Likewise, endemism may not reflect origins, for example if a lineage has gone extinct everywhere else in its range. Finally, a lineage is not technically autochthonous if it is the result of radiation from migrants, such as occurs with founder effects. Regions with complex topography require treatment as islands with island biogeographical theory.

Consider adding to the Results a figure that updates the biogeographic provinces to the authors’ proposed scheme. A nice addition to this figure would be the counts/percentages of the different groups shown within the polygons of the biogeographic provinces.

The niche modeling component is a strength of the paper, and the results provided here make for testable predictions in the near future. The reasoning for the eoclogical niche modeling is well done and there is a clear economic impact.

Specific comments

L51. The reasoning for why Orthoptera are a good model for Anatolian biogeography is not convincing. Surely there are other well known, diverse lineages with direct ties to plant communities as primary consumers, Lepidoptera for example. Consider mentioning restricted dispersal ability for many flighless Orthoptera, which make them particularly useful for reconstructing biogeography, unlike groups like Lepidoptera. Low dispersal also causes behind regional endemism and radiation.

L58. Statement about biogeogrpahical signal in phylogenies has no references.

L113. Fig. 1A. Red region “P.” not defined in caption.

L118. Vague. What is meant by “new definition?” Given the next sentences, is a “definition” possible? If a general classification misses nuances, is dividing Anatolia into biogeographical provinces useful? That statement seems to undermine the present work.

L120. Consider introducing how biogeographical research from other regions with complex geograph and climate, such as southwestern South America or western North America, have met those challenges. Such challenges have not stopped other authors from defining biogeographic regions.

L124. I am not sure what is meant when endemics are said to “requires its own parameterization.” Endemic species are part of any biogeographical scheme.

L172-177. The Zohary work deserves mention in the Introduction, and may serve as a beginning to the Intro as mentioned above. In the next statements this scheme is shown to be similar to previous Orthoptera-centered biogeography studies. There is also an internal conflict with the reasoning in this paper: the Zohary eco-biogeographic work served as the foundation for the Methods of this study, when this approach was earlier criticized in the Intro (L125). The authors must justify why their eco-geographic approach is improved over previous studies using a similar approach. It may simply be that Anatolian endemism is here taken into account where it was not before; if so, please state clearly near L124.

L230-231. Percentage reported twice.

L233. Spelling = Tridactylidae and Tetrigidae for those families. Not consistent with tables e.g. Table 2.

L258. Define these variables in Methods if they are to be reported. Readers may want to see at a glance which variables explain changing distributions.

L266. Thanks to the authors for cautioning the reader as to the interpretation of the modeling result.

L595. Indicate variables source in caption, WorldClim.

L714. These numbers seem impressive. Please define a threshold value for a hotspot. How do these numbers compare with other regions biodiverse for Orthoptera, such as Argentina or California, USA?

L724. Incorrect spelling of families as in L233.

L731. No reference given, this is bold claim regarding evolution, especially without a reference!

L737. A lineage may radiate in an area that is not its (geographic) center of origin. The statements regarding origins are given here without evidence.

L740-741. Autochthonus may not fit the distribution described here. Pamphagidae may have radiated here after migration from another center, invalidating that hypothesis.

L755. Include reasoning in Introduction, as indicated above in L172.

L783. Origin and radiation are non-mutually exclusive hypotheses for the Pamphagidae. Both may not be true, and evidence is not provided here to evaluate them.

L795-800.  These statements are adding new findings to the Discussion. Decisions on biogeographic province boundaries must be justified earlier in the Introduction and Methods, and the outcomes belong in the Results.

Comments on the Quality of English Language

I was quite impressed with the quality of the English. Keep it up!

Author Response

Thank you very much for the constructive comments. All the comments were considered word by word and the manuscript was revised according some comments and explanations provided for the others. 

General comments

The paper by Ciplak and Uluar has three goals: update the list of Anatolian Caelifera, biogeographically classify Caelifera fauna, and predict range changes of economically injurious grasshoppers through ecological niche modeling. The impact of this work may be broadened by inclusion of general biogeography. The Introduction is entirely and narrowly focused on grasshoppers. This is a grasshopper specific work to be sure, but biogeographic provinces are best drawn from shared distributions of diverse life forms. Despite Orthoptera representing a basis for Anatolian biogeography, consider beginning the introdution with background about the biogeograhy of Anatolia in general as defined by the distributions of numerous organisms. The first time the reader sees the vegetation-based provinces that the authors use is in the Methods. Introducing this scheme in the Intro will set context for the historical and current Orthoptera studies, explain how the authors define biogeographical provinces, which is important given the dissimilarities among the provinces as defined by historical authors working on the Orthoptera.

BC- Thanks for the detailed comments! We apply a previously defined eco-geographic classification to Anatolian grasshoppers. This is the reason we have mention Zohary in Methods section. Our paper has no aim to compare different biogeographic classification and to decide about superiority of one over others. The manuscript aims to establish a biogeographic template and to use this template in evaluating pest potential of some species. Considering comments by other two reviewers we have made some revision in Introduction section and we hope these revisions also meet comments here.

It is worth mentioning and referencing the endemism of Orthoptera in other parts of the world and not just in Anatolia. For example, the Orthoptera of California, USA are highly endemic as reported by Harrison 2013 “Plant and Animal Endemism in California.” Orthoptera are generally useful in defining biogeography because they are primary consumers, thus following regional floras rather closely. Grasshoppers also codistribute with soil types, perhaps worth mentioning or considering when detailing biogeographical provinces. This strengthens the choice to work on Orthoptera by illustrating how useful they are for understanding biogeogrpahy worldwide.

BC- Thanks for the detailed comments! We agree that there are many other areas with high and/or endemic biodiversity. If our study would aim to prepare a comparative biogeographic manuscript (both for areas worldwide and for different lineages) it would be necessary to present a text using such a perspective. Respectfully, we prefer to left the perspective in the limits already exist.   

In several places in the manuscript claims are given without evidence, particularly claims regarding origins or evolutionary history. In some cases reference(s) may simply be missing, in others the data may not exist, as presented here the reader cannot evaluate.

I caution the authors to carefully consider alternative biogeographic hypotheses and be clear with their definitions. A lineage may be diverse in a region without that region being its center of origin. Likewise, endemism may not reflect origins, for example if a lineage has gone extinct everywhere else in its range. Finally, a lineage is not technically autochthonous if it is the result of radiation from migrants, such as occurs with founder effects. Regions with complex topography require treatment as islands with island biogeographical theory.

BC- Thanks for the comments in preceding two paragraphs! These comments are also given under two subheadings below (named as specific comments) and we presented our explanations along them too. As explained there, we think there is no statement without evidence! The contemporary range of a lineage and its diversity in this area includes significant evidences about it is centre of origin. If there is no extant and/or fossil representative outside the range area, it is not logical to assume somewhere else out of the range area as centre of origin. For example, genus Glyphotmethis has 13 species/subspecies in Anatolia with single representative in the adjoining Balkan/Thrace. Then, how/why we assume that it has centre of origin e.g. in Africa or North America? The case defined in above two paragraphs may be possible for some rare relict endemics, and as far as we know there is no such Orthoptera reported in the published studies. Respectfully, as explained below also, we do not agree with this definition of centre of origin and prefer not to revise text accordingly.       

Consider adding to the Results a figure that updates the biogeographic provinces to the authors’ proposed scheme. A nice addition to this figure would be the counts/percentages of the different groups shown within the polygons of the biogeographic provinces.

BC- Thanks for the comment! A figure depicting biogeographic provinces is already in the text (Figure 2)!

The niche modeling component is a strength of the paper, and the results provided here make for testable predictions in the near future. The reasoning for the eoclogical niche modeling is well done and there is a clear economic impact.

BC- Thanks! 

Specific comments

L51. The reasoning for why Orthoptera are a good model for Anatolian biogeography is not convincing. Surely there are other well known, diverse lineages with direct ties to plant communities as primary consumers, Lepidoptera for example. Consider mentioning restricted dispersal ability for many flighless Orthoptera, which make them particularly useful for reconstructing biogeography, unlike groups like Lepidoptera. Low dispersal also causes behind regional endemism and radiation.

L58. Statement about biogeogrpahical signal in phylogenies has no references.

BC- Thanks for the comments! The Introduction section is partly revised, considering comments by other reviewers. We should note that we agree with reviewer that there may be other lineages may fit this pattern. However, as stated above a comparative updating of Anatolian biogeography is not among aims of the manuscript (we hope will be able to do in another paper). Further, model position of Orhoptera for biogeography of Anatolia is reported in many other publications, e.g. Uvarov (1921) or Ciplak (2004).   

L113. Fig. 1A. Red region “P.” not defined in caption.

BC- Thanks! Defined!

L118. Vague. What is meant by “new definition?” Given the next sentences, is a “definition” possible? If a general classification misses nuances, is dividing Anatolia into biogeographical provinces useful? That statement seems to undermine the present work.

BC- Thanks! Revised for clarity!

L120. Consider introducing how biogeographical research from other regions with complex geograph and climate, such as southwestern South America or western North America, have met those challenges. Such challenges have not stopped other authors from defining biogeographic regions.

BC- Thanks for the comment! But, we prefer to retain the sentences!

L124. I am not sure what is meant when endemics are said to “requires its own parameterization.” Endemic species are part of any biogeographical scheme.

BC- Thanks! Revised for clarity!

L172-177. The Zohary work deserves mention in the Introduction, and may serve as a beginning to the Intro as mentioned above. In the next statements this scheme is shown to be similar to previous Orthoptera-centered biogeography studies. There is also an internal conflict with the reasoning in this paper: the Zohary eco-biogeographic work served as the foundation for the Methods of this study, when this approach was earlier criticized in the Intro (L125). The authors must justify why their eco-geographic approach is improved over previous studies using a similar approach. It may simply be that Anatolian endemism is here taken into account where it was not before; if so, please state clearly near L124.

BC- Thanks, but we could not grasp the point in this comment! We have not criticized the eco-biogeographic classification by Zohary! But, we mentioned to Zohary in Introduction. 

L230-231. Percentage reported twice.

BC- Thanks! Corrected!

L233. Spelling = Tridactylidae and Tetrigidae for those families. Not consistent with tables e.g. Table 2.

BC- Thanks! Corrected throughout text!

L258. Define these variables in Methods if they are to be reported. Readers may want to see at a glance which variables explain changing distributions.

BC- Thanks! Each defined!

L266. Thanks to the authors for cautioning the reader as to the interpretation of the modeling result.

BC- Thanks!

L595. Indicate variables source in caption, WorldClim.

BC- Thanks! Indicated!

L714. These numbers seem impressive. Please define a threshold value for a hotspot. How do these numbers compare with other regions biodiverse for Orthoptera, such as Argentina or California, USA?

BC- Thanks! Respectfully, we could not grasp why it is necessary to give a threshold! There are several different parts of the world containing a rich biodiversity and the aim of the study is not presenting a comparison!

L724. Incorrect spelling of families as in L233.

BC- Thanks! Corrected!

L731. No reference given, this is bold claim regarding evolution, especially without a reference!

BC- Respectfully, we do not agree with the comment. If a species is endemic to a geographic area with 95% likelihood it evolved there. The term relict endemic which are once widespread and later become endemic to somewhere are rare! We do not know any such orthopteran species. So, we think there is no need for further references! Also, supporting references are given.

L737. A lineage may radiate in an area that is not its (geographic) center of origin. The statements regarding origins are given here without evidence.

BC- Respectfully, again we do not agree with the comment. If a genus includes for example 10 species and if all occurs in particular area, that area is centre of origin and the lineage radiated in that area. The term relict endemic which are once widespread and later become endemic to a restricted are rare! We do not know any such orthopteran genus or species that have origin place far of its contemporary distribution area. Several genera of Pamphagidae are Anatolian in distribution and they have no representatives (either extant or fossil) in other areas.

L740-741. Autochthonus may not fit the distribution described here. Pamphagidae may have radiated here after migration from another center, invalidating that hypothesis.

BC- Respectfully, we do not agree with the comment (explained above).

L755. Include reasoning in Introduction, as indicated above in L172.

BC- Respectfully, explained above!

L783. Origin and radiation are non-mutually exclusive hypotheses for the Pamphagidae. Both may not be true, and evidence is not provided here to evaluate them.

BC- Respectfully, we do not agree with the comment (explained above).

L795-800.  These statements are adding new findings to the Discussion. Decisions on biogeographic province boundaries must be justified earlier in the Introduction and Methods, and the outcomes belong in the Results.

BC- Respectfully, explained above and along general comments!